# Prediction of Urban Forest Aboveground Carbon Using Machine Learning Based on Landsat 8 and Sentinel-2: A Case Study of Shanghai, China

Huimian Li [1,2,3], Guilian Zhang [4,5], Qicheng Zhong [4,5], Luqi Xing [4,5] and Huaqiang Du [1,2,3,*]

1. State Key Laboratory of Subtropical Silviculture, Zhejiang A&F University, Hangzhou 311300, China
2. Key Laboratory of Carbon Cycling in Forest Ecosystems and Carbon Sequestration of Zhejiang Province, Zhejiang A&F University, Hangzhou 311300, China
3. School of Environmental and Resources Science, Zhejiang A&F University, Hangzhou 311300, China
4. Shanghai Academy of Landscape Architecture Science and Planning, Shanghai 200232, China
5. Shanghai Engineering Research Center of Landscaping on Challenging Urban Sites, Shanghai 200232, China
* Correspondence: duhuaqiang@zafu.edu.cn

**Abstract:** The aboveground carbon storage (AGC) of urban forests is an important indicator reflecting the ecological function of urban forests. It is essential to monitor the AGC of urban forests and analyze their spatiotemporal distributions. Remote sensing is a technical tool that can be leveraged to accurately monitor forest AGC, whereas machine learning is an important algorithm for the accurate prediction of AGC. Therefore, in this study, single Landsat 8 (L) remote sensing data, single Sentinel-2 (S) remote sensing data, and combined Landsat 8 and Sentinel-2 (L + S) data are used as data sources. Four machine learning methods, support vector regression (SVR), random forest (RF), XGBoost (extreme gradient boosting), and CatBoost (categorical boosting), are used to predict forest AGC based on two phases of forest sample plots in Shanghai. We chose the optimal model to predict the AGC and simulate the spatiotemporal distribution. The study shows that both machine learning models based on separate Landsat 8 OLI and Sentinel-2 satellite remote sensing data can accurately predict the AGC and spatiotemporal distribution of the Shanghai urban forest. Nevertheless, the accuracy of the combined data (L + S) and CatBoost-integrated AGC models is higher than the others, with fitting and validation accuracy $R^2$ values of 0.99 and 0.70, respectively. The RMSE was also smaller at 0.67 and 6.29 Mg/ha, respectively. The uncertainty of the AGC spatial distribution in the Shanghai urban forest derived from the CatBoost model prediction from the 2016–2019 data was small and consistent with the actual situation. Furthermore, the statistics showed that the AGC of the Shanghai forest increased from 24.90 Mg/ha in 2016 to 25.61 Mg/ha in 2019.

**Keywords:** urban forest; AGC; remote sensing; machine learning

## 1. Introduction

Cities cover less than 1% of the Earth's surface but account for 71% of global $CO_2$ emissions, and the International Energy Agency predicts that this proportion will grow to 76% by 2030 [1]. Therefore, the issue of urban $CO_2$ emissions has become the focus of global carbon emission reduction and low-carbon development [2]. Aboveground carbon storage (AGC) is an important indicator reflecting the $CO_2$ absorption capacity of urban forests and evaluating the quality of those ecosystems [3]. The urban forest is a green space system consisting of forest patches, forest strips, and scattered trees contained within urban areas [4]. With the gradual expansion of urban environments, many of the functions offered by urban forests, such as $CO_2$ emission reduction, air purification, PM2.5 absorption, urban rainfall, flood control, water quality improvement, noise reduction, and microclimate improvement, have received increasing attention [5]. Therefore, the study of urban forest AGC and its spatial and temporal distribution is of great significance to the construction

of forested and low-carbon cities, which is a popular focus of domestic and international research [6,7].

Currently, the methods of estimating forest AGC generally include field surveys, model simulations, and remote sensing inversions. Considering field surveys, the inventorying of sample plots uses two types of survey data (diameter at breast and tree height) to calculate carbon stocks through the anisotropic growth equation. The estimated value obtained, although accurate, requires considerable human and material resources and a lengthy survey period, cannot wholly reflect the spatial distribution pattern of the whole region, and has certain destructive properties [8]. Model simulation methods, such as BIOME-BGC, mainly estimate carbon stocks by simulating physiological–ecological processes such as photosynthesis, respiration, and decomposition in ecosystems [9]. It should be noted that various vegetation parameters are needed to effectively simulate forest carbon stocks; thus, when the available input parameters are inadequate or missing due to acquisition difficulty, the prediction results will be substantially impacted [10]. The remote sensing estimation method mainly establishes a complete mathematical model, and its analytical formula incorporates the information received from satellites and directly measured biomass. Finally, this approach uses an analytical formula to estimate the forest biomass in other areas [11]. Remote sensing data offers the advantages of fast acquisition and high temporal resolutions and can cover large spatial scales [12]. As remote sensing technology continues to rapidly develop, the high-resolution images and multispectral images obtained by various satellites can collect more detailed vegetation information, such as texture features, and be used for vegetation indices [13,14]. Additionally, these datasets can be combined with the growing suite of machine learning methods, such as the random forest algorithm, support vector regression, and deep learning techniques [15]. Therefore, forest carbon stock estimation using remote sensing technology combined with ground survey data is more accurate [16].

Scholars have conducted substantial research on the remote sensing-based estimations of urban forest carbon stocks [17]. The commonly used multiple linear regression models have some advantages in predicting forest biomass [18], but linear regression cannot fully resolve the complex relationship between independent variables and AGC. In recent years, the emergence of machine learning algorithms, such as artificial neural networks (NN), support vector regression (SVR), random forest (RF), deep learning (DL), and ensemble learning (EL), has greatly improved the accuracy of forest AGC estimation [19]. Zou analyzed the application of a multiple linear regression model, logistic regression model, and neural network model in estimating the urban forest carbon stock in Shenzhen City using remote sensing images from Landsat 8 as the data source. It showed that the neural network model has the highest estimation accuracy [20]. Zhang used Landsat TM and OLI satellite remote sensing data to estimate the spatial and temporal distribution of urban forest carbon stocks in the Hangzhou–Jiahu region using RF [21]. Dong et al. used Worldview-2 high-resolution remote sensing data as the data source and DL to achieve a highly accurate estimation of AGC in the Leizhu forest in Lin'an District, Hangzhou [22]. EL is a branch of machine learning that improves model stability, learning accuracy, and generalization capability by integrating and combining multiple learners, including boosting, bagging, and stacking algorithms, to accomplish learning tasks [23]. The boosting algorithm, also known as the augmented learning or boosting method, improves the model's accuracy by transforming weak learning rules into strong learning rules [24]. Among the boosting algorithms, two methods, XGBoost (extreme gradient boosting) and CatBoost (categorical boosting), are again typical representatives, and they are preferred for cases with small training samples. Scholars have obtained better results using both methods to estimate forest AGC [25].

In summary, monitoring the AGC of urban forests is important for evaluating the function of urban forest carbon sinks and low-carbon city construction. Remote sensing is a technical means to accurately monitor urban forest AGC, and machine learning is an important algorithm that can be applied to improve the accuracy of remote sensing-based

AGC estimation. The purpose of this study is to accurately simulate urban forest AGC based on machine learning models and multi-source remote sensing data and analyze their spatiotemporal distributions. This study takes the Shanghai urban forest as the research object; uses two kinds of remote sensing data, Landsat 8 OLI and Sentinel-2, as the data sources; and extracts remote sensing variables and metrics, such as the vegetation index and texture, and combines them with sample plot survey data. Four machine learning methods, SVR, RF, XGBoost, and CatBoost, are then used to construct the Shanghai urban forest AGC estimation model from the remotely sensed data. Based on the model accuracy evaluation, the model with the best performance and strong generalization ability is selected to estimate the carbon stock and spatial and temporal distribution of the Shanghai urban forest. The study's results will provide an important reference for evaluating the carbon sink capacity of Shanghai's urban forest and its role in constructing a low-carbon city.

## 2. Materials and Methods

### 2.1. Study Area

Shanghai is one of China's most urbanized and fastest growing cosmopolitan cities. Shanghai is located between 120°50′ E and 121°53′ E and between 30°40′ N and 31°50′ N. It is situated on the coast of the East China Sea and the Yangtze River Delta, as shown in Figure 1. The city of Shanghai covers an area of 6340.5 square kilometers, of which Huangpu, Pudong New Area, Xuhui, Changning, Putuo, Jing'an, Hongkou, Yangpu, and other main urban areas cover an area of 977.1 square kilometers. Baoshan, Jinshan, Jiading, Minhang, Fengxian, Qingpu, Songjiang, Chongming, and other suburban areas cover an area of 5363.4 square kilometers [26]. By the end of 2020, the forested area of Shanghai reached 1758 million acres, and the forest coverage rate reached 18.49%. The vegetation along urban streets, parks, and suburban woodlands is dominated by lady's mantle (*Ligustrum lucidum*), magnolia *(Magnolia grandiflora)*, balsam fir *(Cinnamomum camphora)*, *Populus*, *Cedrus deodara*, and other subtropical evergreens, deciduous broadleaf, and evergreen broadleaf species [27].

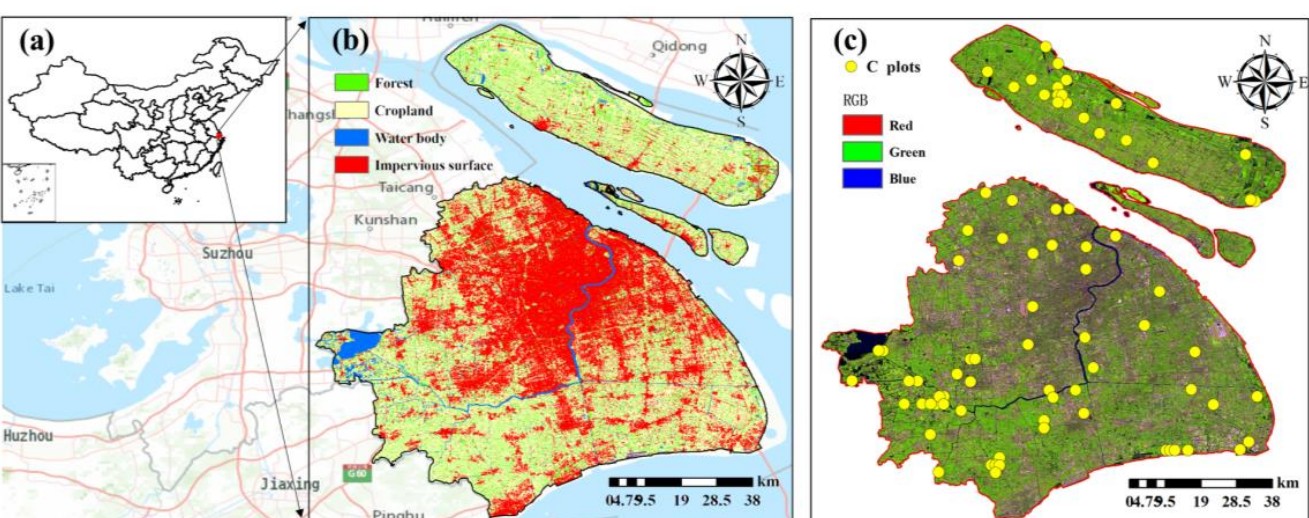

**Figure 1.** (**a**) China's border; (**b**) classified data of Shanghai in 2019; (**c**) Landsat 8 image of Shanghai province and forest aboveground carbon (AGC) plots in 2019.

### 2.2. Datasets and Processing

#### 2.2.1. Processing Observed Data

The ground sample plot data were obtained from 81 urban forest fixed sample plots in Shanghai that were each 25.8 m × 25.8 m in size. The distribution of the sample plots is shown in Figure 1. The 81 sample plots were surveyed over a period of 4 years, and each sample plot was surveyed twice. The 40 sample plots were surveyed in the summer of 2016

and 2018, respectively; the remaining 41 sample points were surveyed in the summer of 2017 and 2019. In this study, the remote sensing data corresponding to the time of sample points are selected for the spatiotemporal estimation of urban forest AGC.

During the field survey, tree height, diameter at breast height, height below the canopy, and crown width were recorded for trees with a diameter at breast height greater than 5 cm. The AGC of individual trees was calculated by combining the anisotropic growth equations [28] of different tree species to statistically derive the total AGC of each site. As outlier AGC values will significantly damage model training, this study used the double standard deviation method [29] to ensure the observed AGCs were within 95% confidence, and 103 sample plots were selected for estimating urban forests AGC. The AGC statistics for the 103 samples are shown in Table 1. These sample plots were divided into two parts using a 3:1 ratio, of which 75% of the samples were used for model construction, whereas the other 25% were used for model accuracy assessment.

**Table 1.** Summary of the forest AGC plots of Shanghai.

| ID | Year | Sample Dimension | Min (Mg/ha) | Max (Mg/ha) | Mean (Mg/ha) | SD (Mg/ha) |
|----|------|------------------|-------------|-------------|--------------|------------|
| 1 | 2016 | 27 | 1.71 | 53.72 | 26.86 | 12.87 |
|   | 2017 | 24 | 2.98 | 55.54 | 26.55 | 11.82 |
| 2 | 2018 | 26 | 2.55 | 52.63 | 28.05 | 11.4 |
|   | 2019 | 26 | 3.89 | 52.33 | 29.91 | 12.57 |

### 2.2.2. Landsat 8 and Sentinel-2 Remote Sensing Data

Landsat 8 is an imaging mission with large time span, good data quality, and high resolution collected by National Aeronautics and Space Administration (NASA) and the United States Geological Survey (USGS). It is equipped with an operational land imager (OLI) to monitor and help manage the use of agricultural, forestry, animal husbandry, and water resources and investigate and forecast various serious natural disasters (such as earthquakes) and environmental pollution. Landsat 8 data can be downloaded for free (https://earthexplorer.usgs.gov/ (accessed on 1 August 2021)).

Sentinel-2 is a large-scale, high-resolution, multispectral imaging mission funded by the European Union, European Space Agency (ESA), and Copernicus Programme, which supports Copernicus land monitoring and research, including vegetation, soil, and water cover monitoring and observation of inland waterways and coastal areas. The data of Sentinel-2 can be obtained from scihub (https://scihub.copernicus.eu/ (accessed on 1 June 2022)).

### 2.2.3. Remote Sensing Data Preprocessing

As shown in Table 2, in this study, remote sensing data from two sources, Landsat 8 OLI and Sentinel-2 were selected to estimate the AGC of the Shanghai urban forest based on the ground sample survey results. Satellite data were selected from 2016 to 2019, and only imagery with low cloud cover was used to ensure data quality. Two scenes from the Landsat 8 image data over Shanghai, with strip numbers 118/039 and 118/040, were used to consider the influence of the atmosphere, aerosols, and other factors in the image acquisition process. This study performed radiometric calibration, FLAASH atmospheric correction, and geometric correction on the Landsat 8 image data [30–32] and stitched and cropped the corrected data.

**Table 2.** Acquisition date and cloud coverage (C) (%) of the images.

| Satellite | Data ID | 2016 | | Data ID | 2017 | |
| --- | --- | --- | --- | --- | --- | --- |
| | | Date | Cloud | | Date | Cloud |
| Landsat 8 | LC81180382016202LGN00 | 20/07/2016 | 6.09 | LC81180382017236LGN00 | 24/08/2017 | 0.40 |
| | LC81180392016122LGN00 | 01/05/2016 | 1.25 | LC81180392017236LGN00 | 24/08/2017 | 0.26 |
| Sentinel-2 | N0202_R089_T51RTQ | 04/05/2016 | 2.62 | N0205_R089_T51RTQ | 28/07/2017 | 0.13 |
| | N0202_R089_T51RUQ | 04/05/2016 | 5.17 | N0205_R046_T51RUQ | 04/08/2017 | 0.56 |
| | N0204_R046_T51SUR | 30/06/2016 | 15.79 | N0205_R089_T51SUR | 27/08/2017 | 1.15 |
| | N0204_R089_T51SUR | 02/08/2016 | 19.70 | N0205_R089_T51RUQ | 28/07/2017 | 0.58 |
| | N0204_R046_T51RUQ | 30/06/2016 | 4.63 | N0205_R046_T51RUQ | 26/05/2017 | 0.05 |
| Satellite | Data ID | 2018 | | Data ID | 2019 | |
| | | Date | Cloud | | Date | Cloud |
| Landsat 8 | LC81180382018143LGN00 | 23/05/2018 | 18.43 | LC81180382019210LGN00 | 29/07/2019 | 10.78 |
| | LC81180392018143LGN00 | 23/05/2016 | 4.37 | LC81180392019210LGN00 | 29/07/2019 | 1.66 |
| Sentinel-2 | N0206_R089_T51RTQ | 04/05/2018 | 0.06 | N0208_R089_T51RTQ | 17/08/2019 | 0.01 |
| | N0206_R089_T51RUQ | 04/05/2018 | 0.03 | N0208_R089_T51RUQ | 17/08/2019 | 0.15 |
| | N0204_R046_T51SUR | 04/05/2018 | 15.40 | N0208_R089_T51SUR | 17/08/2019 | 0.09 |
| | N0206_R089_T51SUR | 13/06/2016 | 15.92 | N0208_R046_T51RUQ | 14/08/2019 | 2.62 |
| | N0206_R046_T51RUQ | 29/08/2016 | 4.10 | N0207_R046_T51RUQ | 15/08/2019 | 1.01 |

Sentinel-2 carries a spectral imager that spans 13 spectral bands. These include four bands with a spatial resolution of 10 m: the blue band (490 nm), the green band (560 nm), the red band (665 nm), and the near-infrared band (842 nm); four bands with a spatial resolution of 20 m, which mainly consists of the four bands used to characterize the red spectrum of vegetation (705, 740, 783, and 865 nm); two shortwave infrared bands (1610 and 2190 nm) for snow and ice detection; and three bands with a resolution of 60 m that are mainly used for atmospheric correction, etc. [33]. In this study, bands with spatial resolutions of 10 and 20 m are mainly used. For Sentinel-2 Level-1C images, atmospheric correction was performed using the Sen2cor tool (http://step.esa.int/main/snap-supported-plugins/sen2cor/ (accessed on 1 June 2022)) [34]. In addition, to be consistent with Landsat 8, the spatial resolution was resampled to 30 m using the nearest neighbor method [35].

## 3. Research Methodology

### 3.1. Remote Sensing Variable Settings

As shown in Table 3, the remote sensing variables used in this study, such as the vegetation indices and texture features, were derived from the original image and composite bands. The vegetation indices include the difference vegetation index (DVI), normalized difference vegetation index (NDVI), normalized difference water index (NDWI), enhanced vegetation index (EVI), and ratio vegetation index (RVI) [36]. NDVI is the most commonly used vegetation index for the qualitative and quantitative evaluation of vegetation coverage and its growth vitality [37]. NDWI is a normalized water index based on the green and near-infrared bands. EVI has a narrower range for red light and near-infrared detection bands, which can detect sparse vegetation and reduce the impact of water vapor. RVI can better reflect the differences in vegetation coverage and growth status. DVI can detect vegetation growth status and coverage and eliminate some radiation errors [38].

The texture features are based on the original image waveform. The texture features (i.e., mean, variance, homogeneity, contrast, dissimilarity, entropy, angular second-order moment, and correlation) are based on the original image bands calculated using the grayscale co-occurrence matrix [39]. The window size for texture feature extraction is set to $3 \times 3$, $5 \times 5$, $7 \times 7$, $9 \times 9$, and $11 \times 11$.

**Table 3.** Information on remote sensing-derived variables.

| Type | Name | Calculation Models or Descriptions | Abbreviation | Remarks |
|---|---|---|---|---|
| Landsat Original Band | Coastal | Band 1 | B1 | |
| | Blue | Band 2 | B2 | |
| | Green | Band 3 | B3 | |
| | Red | Band 4 | B4 | Landsat 8 OLI data |
| | NIR | Band 5 | B5 | |
| | Swir1 | Band 6 | B6 | |
| | Swir2 | Band 7 | B7 | |
| Sentinel-2 Original Band | Blue | Band 2 | S_B1 | |
| | Green | Band 3 | S_B2 | |
| | Red | Band 4 | S_B3 | |
| | Red-edge 1 | Band 5 | S_B4 | |
| | Red-edge 2 | Band 6 | S_B5 | |
| | Red-edge 3 | Band 7 | S_B6 | Sentinel-2 data |
| | NIR1 | Band 8 | S_B7 | |
| | NIR2 | Band 8A | S_B8 | |
| | Swir1 | Band 9 | S_B9 | |
| | Swir2 | Band 10 | S_B10 | |
| Vegetation Index | NDVI | $(\mathrm{NIR} - \mathrm{R}) / (\mathrm{NIR} + \mathrm{R})$ | NDVI | |
| | NDWI | $(\mathrm{G} - \mathrm{NIR}) / (\mathrm{G} + \mathrm{NIR})$ | NDWI | |
| | EVI | $2.5\,(\mathrm{NIR} - \mathrm{R}) / (\mathrm{NIR} + 6\mathrm{R} - 7.5\mathrm{B} + \mathrm{L})$ | EVI | L takes value for 0.5 [40] |
| | RVI | $\mathrm{NIR}\,/\,\mathrm{R}$ | RVI | |
| | DVI | $\mathrm{NIR} - \mathrm{R}$ | DVI | |
| Texture [10] | Mean | $\sum_{i=0}^{N-1}\sum_{j=0}^{N-1} i P(i,j)$ | MEA | $P(i,j) = \frac{V(i,j)}{\sum_{i=0}^{N-1}\sum_{j=0}^{N-1} V(i,j)}$, where $V(i,j)$ is the ith row of the jth column in the Nth moving window; |
| | Variance | $\sum_{i=0}^{N-1}\sum_{j=0}^{N-1} (i - mean)^2 P(i,j)$ | VAR | |
| | Homogeneity | $\sum_{i=0}^{N-1}\sum_{j=0}^{N-1} \frac{P(i,j)}{1+(i-j)^2}$ | HOM | $u_{x=} \sum_{j=0}^{N-1} j \sum_{i=0}^{N-1} P(i,j)$ |
| | Contrast | $\sum_{|i-j|=0}^{N-1} |i-j|^2 \left\{ \sum_{i=1}^{N}\sum_{j=1}^{N} P(i,j) \right\}$ | CON | $u_{y=} \sum_{i=0}^{N-1} i \sum_{j=0}^{N-1} P(i,j)$ |
| | Dissimilarity | $\sum_{|i-j|=0}^{N-1} |i-j| \left\{ \sum_{i=1}^{N}\sum_{j=1}^{N} P(i,j) \right\}$ | DIS | $\sigma_{x=} \sum_{j=0}^{N-1} (j - u_i)^2 \sum_{i=0}^{N-1} P(i,j)$ |
| | Entropy | $-\sum_{i=0}^{N-1}\sum_{j=0}^{N-1} P(i,j) \log(P(i,j))$ | ENT | $\sigma_{y=} \sum_{i=0}^{N-1} (i - u_j)^2 \sum_{j=0}^{N-1} P(i,j)$ |
| | Angular second moment | $\sum_{i=0}^{N-1}\sum_{j=0}^{N-1} P(i,j)^2$ | ASM | |
| | Correlation | $\frac{\sum_{i=0}^{N-1}\sum_{j=0}^{N-1} P(i,j)^2 - \mu_x \mu_y}{\sigma_x \sigma_y}$ | COR | |

The remote sensing-derived variables based on Landsat data include 7 original bands, 5 vegetation indices, and 280 texture features (extracted using 5 window sizes generated from the 7 original bands), for a total of 292 variables. The remote sensing-derived variables based on Sentinel-2 data include 10 original bands, 5 vegetation indices, and 400 texture features (also extracted by 5 kinds of windows generated based on 10 original bands), for a total of 415 variables.

### 3.2. Feature Variable Selection

In machine learning algorithms, selecting feature variables involved in model construction is extremely important to improve the model's performance [41]. The Boruta algorithm proposed by Kursa [42] can select feature variables efficiently, and its goal is to filter all features relevant to the dependent variable so that the influence of feature variables on the dependent variable can be better explained. The core idea of the Boruta algorithm is

to disrupt the original feature order to construct shadow features randomly. The shadow features are trained as a new feature matrix together with the original features. The importance score of the shadow features is used as a benchmark to find all the features related to the dependent variable from the original features through multiple iterations to obtain the optimal features. Obviously, the Boruta algorithm is different from the traditional method of selecting feature variables [43]; for example, it is the least-cost function [44] because the features extracted through the least-cost function do not take into account the nonlinear correlation between the dependent variables. Therefore, the Boruta algorithm is used in this study to filter feature variables from the remote sensing-derived variables discussed in Section 3.1 for AGC model construction.

### 3.3. AGC model Construction Scheme and Method

Based on the variable selection, this study constructs AGC models with three schemes, namely, Landsat 8 remote sensing data (L), Sentinel-2 (S), and Landsat 8 combined with Sentinel-2 (L + S). Four machine learning methods, SVR, RF, XGBoost, and CatBoost are used for modeling. Finally, based on the model accuracy evaluation, the model with good performance and strong robustness is selected to estimate the carbon stock and spatial and temporal distribution of the Shanghai urban forest.

SVR constructs the model by finding a linear regression equation to fit all sample points so that the total variance of sample points from the hyperplane is minimized. SVR maps the nonlinearly separable training samples in the low-dimensional input space to the high-dimensional space by introducing a kernel function to make them linearly separable [45]; thus, the model has good generalization performance and is not easily overfitted [46].

The RF algorithm is an algorithm based on decision tree improvement [47]. The method first forms N sets of samples for the dataset using self-service sampling techniques (bootstrap) [48]. Then, a decision tree model is built for each set of samples separately to form N regression trees. Finally, the average of the results of the N regression trees is used as the predicted value of the carbon stock. RF considers only some remote sensing variables at each split point; thus, the weakly correlated variables have more opportunities to participate in regression tree construction, which increases the reliability of the model. Compared with the traditional multiple linear regression model, RF can better handle the complex covariance between remote sensing variables [49].

The XGBoost algorithm starts from the root node and splits one leaf node at a time, and the optimal split is selected for each possible division. The model is obtained by improving on the gradient boosting decision tree (GBDT) algorithm, which is an integrated algorithm for implementing decision trees as the base classifier and usually consists of multiple decision trees. Nevertheless, it only prunes after the decision tree is constructed. Furthermore, XGBoost adds a regular term in the decision tree construction stage to reduce the model's overfitting, thus improving the model's generalization ability [50,51].

The CatBoost algorithm is also an improved ensemble learning method developed in the framework of the GBDT algorithm, but it implements algorithm integration with a symmetric decision tree as the base learner, uses the same features for splitting at each layer during the operation, and calculates the leaf node values by minimizing the sample loss on the leaf nodes [52]. The CatBoost algorithm has fewer parameters and can efficiently and reasonably handle category-based features [53], whereas CatBoost is able to handle discrete feature data when calculating subset residuals automatically. Therefore, the CatBoost algorithm is highly adaptable to regression problems with multiple input features and noisy samples [54].

### 3.4. Model Accuracy Evaluation Method

In this study, the coefficient of determination ($R^2$), root mean square error (RMSE), and relative root mean square error (rRMSE) are used to evaluate the accuracy of the

model [55,56]. Generally, a higher $R^2$ and lower RMSE and rRMSE indicate that the model has better performance. These can be calculated by:

$$R^2 = \frac{\sum_{i=1}^n (\hat{y}_i - \overline{y}_i)^2}{\sum_{i=1}^n (y_i - \overline{y}_i)^2} \tag{1}$$

$$RMSE = \sqrt{\frac{\sum_{i=1}^n (y_i - \hat{y}_i)^2}{n}} \tag{2}$$

$$rRMSE = \frac{RMSE}{\overline{y}_i} \times 100\% \tag{3}$$

where $y_i$, $\overline{y}_i$, and $\hat{y}_i$ are the measured AGC, mean AGC, and model-predicted AGC, respectively, and n is the number of samples. The closer the $R^2$ value is to 1, the higher the fitting degree. The smaller the RMSE value is, the smaller the dispersion between the true value and the model predicted value.

The technical route of this study is shown in Figure 2.

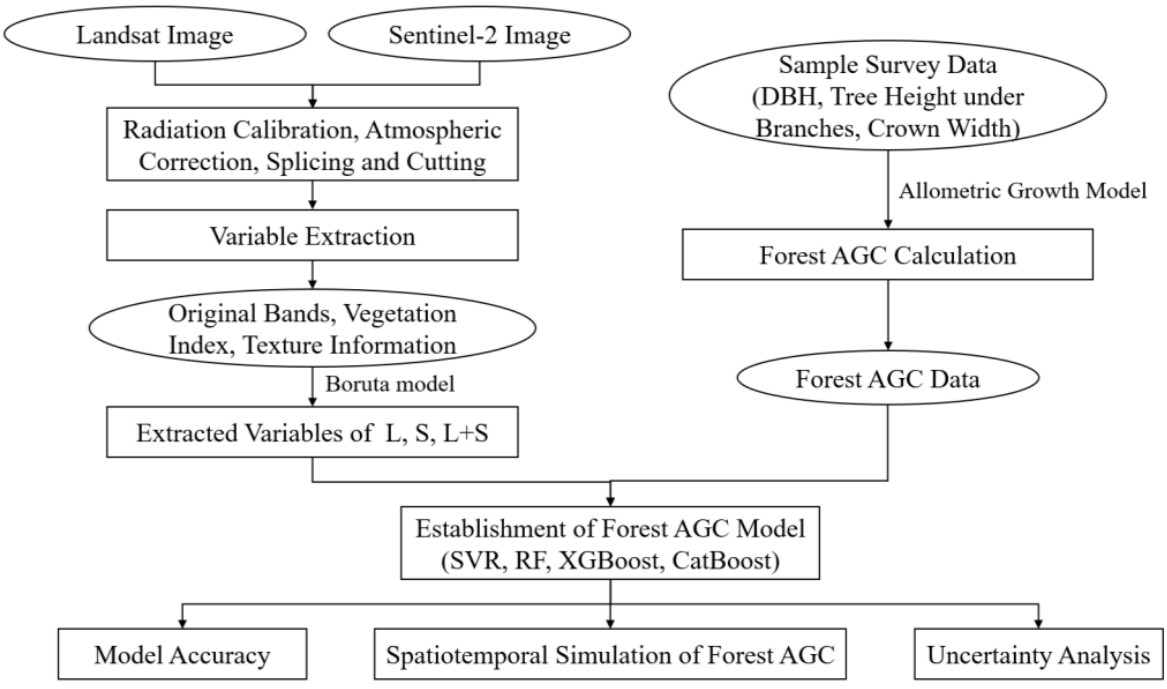

**Figure 2.** Flowchart of steps used in our study.

## 4. Results and Analysis

### 4.1. Variable Screening Results and Importance Analysis

The explanatory power of the selected feature variables for AGC is given in Figure 3, along with a comparison of the three feature variable screening methods, stepwise regression analysis (SR) [57], and recursive feature elimination (RFE) [58]. As seen in Figure 3, the feature variables obtained by the Boruta-based algorithm correlate better with AGC in general, especially for both L and L + S data. Table 4 shows the feature variables screened from the L, S, and L + S data using the Boruta method. Among the three datasets shown in Table 4, texture information has the strongest influence on the AGC. Eleven variables screened in the L dataset are texture information, and among the five feature variables screened in the S dataset, only EVI is not texture information. All variables screened in the L + S dataset are texture information, which consists of 12 variables from the L data and 2 from the S dataset.

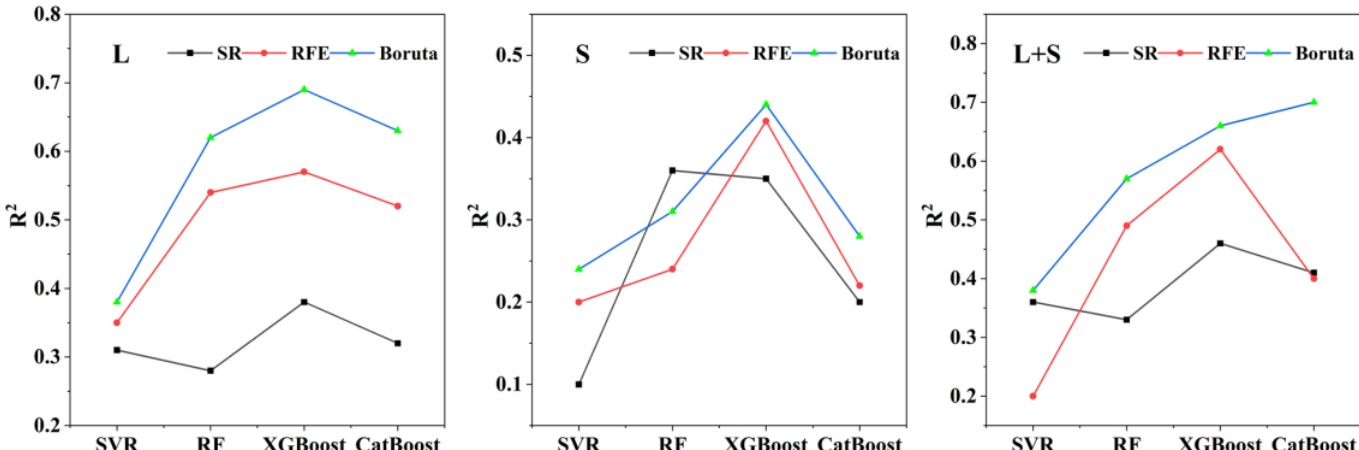

**Figure 3.** The line chart of R2 based on three different feature selection methods and three different data combinations based on four modeling methods.

**Table 4.** Feature variable selection result based on the Boruta method.

| Dataset | Number of Selected Variables | Name of Selected Variables |
|---|---|---|
| L | 11 | W3B4COR, W5B2COR, W5B5VAR, W7B5CON, W7B6VAR, W7B6CON, W9B5CON, W9B5DIS, W11B1CON, W11B5CON, W11B5DIS |
| S | 5 | S_EVI, S_W7B8CON, S_W9B3CON, S_W11B3CON, S_W11B7ENT |
| L + S | 14 | W3B4COR, W5B2COR, W5B5VAR, W7B5CON, W7B6VAR, W9B5CON, W9B5DIS, W11B1CON, W11B5HOM, W11B5CON, W11B5DIS, W11B5ENT, S_W7B5VAR, S_W9B8CON |

Note: WiBjxx refers to the texture information whose window size of the jth band of the image is I; xx refers to MEA, VAR, HOM, CON, DIS, ENT, ASE, and COR; S_ represents Sentinel-2 data; otherwise, it is Landsat 8 data.

*4.2. AGC Model Construction and Prediction Results*

4.2.1. Landsat 8-Based AGC Model and Prediction Results

The accuracy evaluation of the AGC models (SVR, RF, XGBoost, and CatBoost) constructed based on Landsat feature variables are shown in Figure 4. The accuracy $R^2$ in the modeling and validation stages ranges from 0.78 to 0.99 and 0.38 to 0.66, respectively. In addition, the RMSE ranges from 1.43 to 6.49 Mg/ha.

As observed in Figure 4, the overall modeling performance of RF is better than SVR, as the accuracies of the AGC model on RF in the modeling stage (0.94) and validation stage (0.62) outperform that of the SVR model by 20.51 and 63.16%, respectively. The AGC model built by XGBoost shows slightly higher accuracy than that of CatBoost in both the modeling and verification phases.

In general, the predicted urban forest AGC was more reliable with the XGBoost algorithm than the SVR, RF, and CatBoost algorithms and outperforms other models on the accuracies of both the training and verification stages (by 26.92 and 73.68% for SVR, 5.32 and 6.45% for RF, and 2.06 and 4.76% for CatBoost).

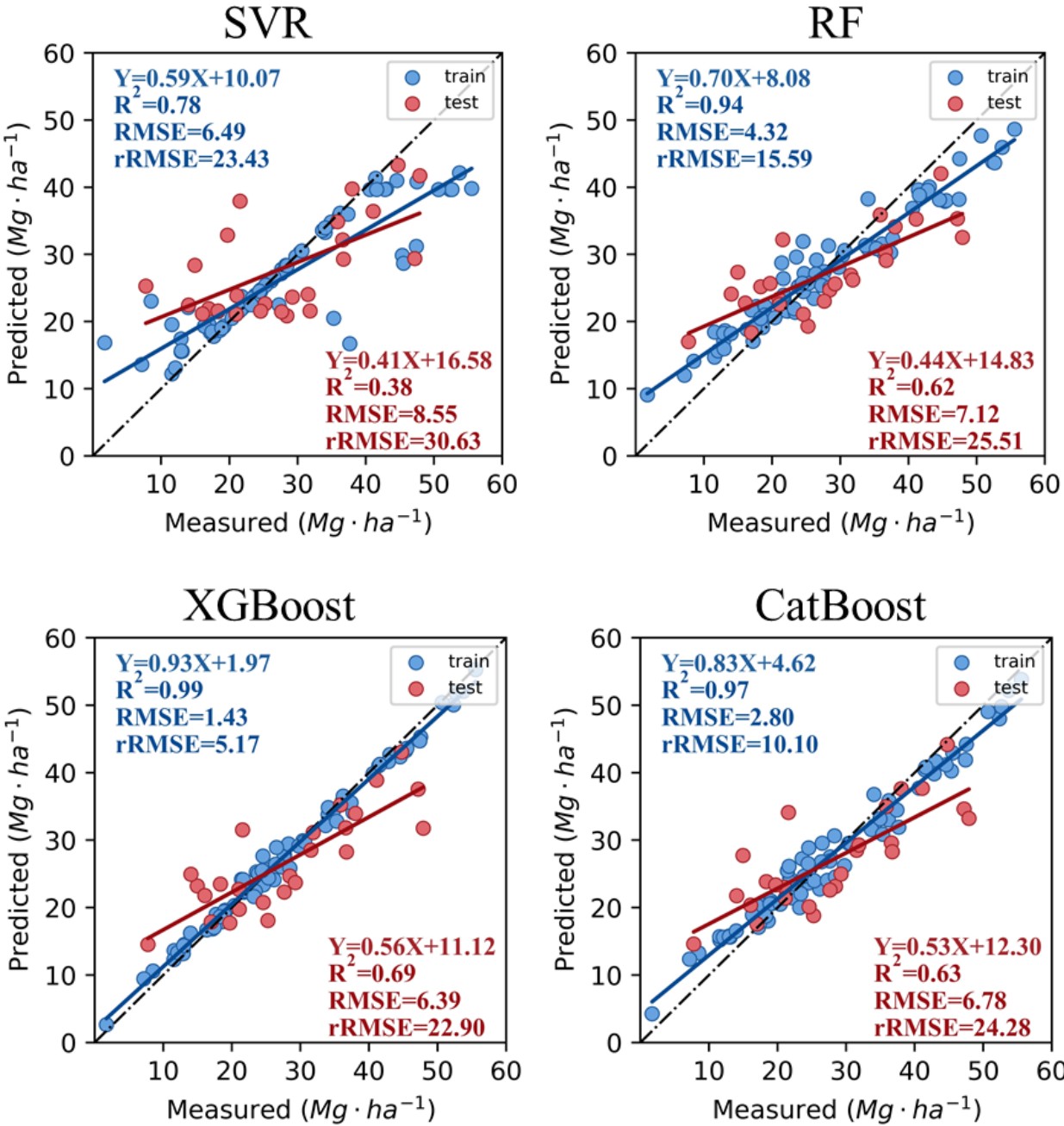

**Figure 4.** Accuracy evaluation of AGC prediction based on Landsat data using four machine learning models.

4.2.2. Sentinel-2-Based AGC Model and Prediction Results

The results of the AGC models based on Sentinel-2 feature variables are shown in Figure 5. The modeling accuracy $R^2$, the validation accuracy $R^2$, and the RMSE of the four models ranged from 0.51 to 0.94, 0.24 to 0.44, and 4.12 to 9.27 Mg/ha, respectively. In Figure 5, the overall modeling performance of RF is better than that of SVR, as the accuracy of the RF model, which is 0.94 in the modeling phase and 0.31 in the validation phase, is higher than the accuracy of the SVR model by 42.42 and 29.17%, respectively, and the RMSE decreases by 3.59 Mg/ha. The accuracy of the XGBoost-based AGC model is greater than that of CatBoost in the modeling and verification phases, and the overall modeling performance of the XGBoost algorithm is slightly better than that of the CatBoost algorithm.

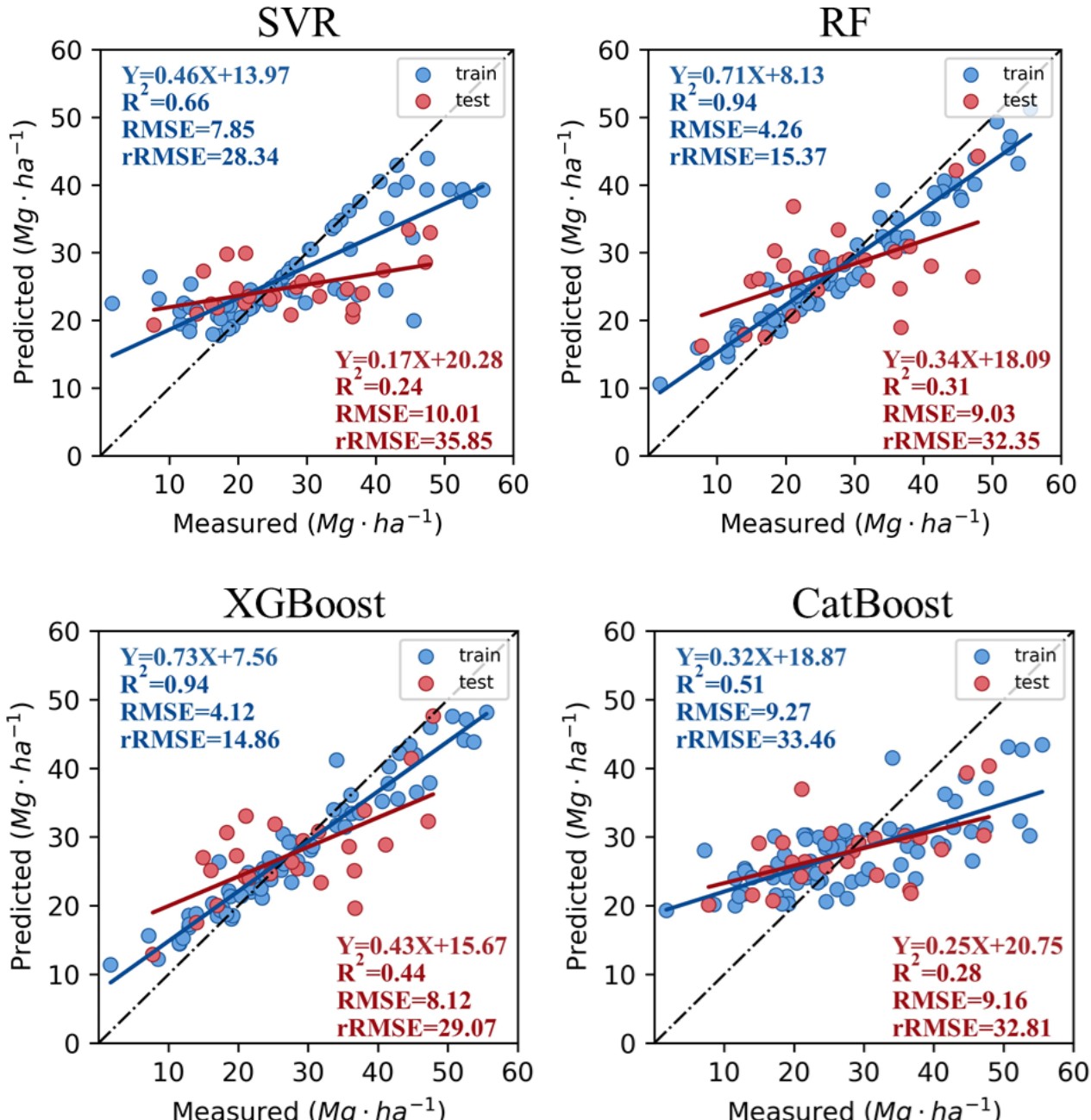

**Figure 5.** Accuracy evaluation of AGC prediction based on Sentinel-2 using four machine learning models.

The modeling accuracy of the XGBoost algorithm is the same as that of RF, but the validation accuracy is improved by 41.94%, and the accuracy of the XGBoost algorithm compared with SVR and CatBoost in the modeling and validation stages is improved by 42.42 and 83.33%, and 84.31% and 57.14%; RMSE decreased by 3.73 Mg/ha and 5.15 Mg/ha; and the AGC model accuracy based on the XGBoost algorithm was the highest.

### 4.2.3. Landsat 8 Combined with the Sentinel-2 AGC Model and Prediction Results

The accuracy of the AGC models constructed using SVR, RF, XGBoost, and CatBoost based on Landsat 8 combined with Sentinel-2's remote sensing-derived variables are shown in Figure 6, with modeling accuracy $R^2$ values between 0.88 and 0.99, validation accuracy $R^2$ values between 0.39 and 0.70, and RMSEs between 0.67 and 5.76 Mg/ha for all four models.

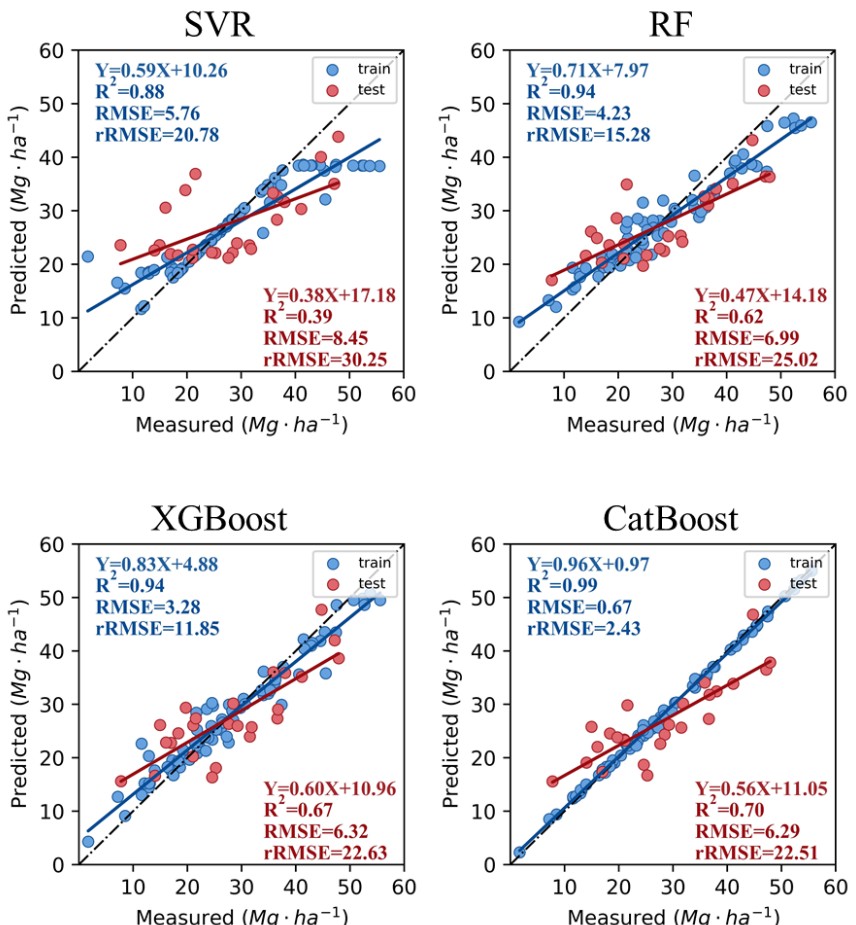

**Figure 6.** Accuracy evaluation of AGC prediction based on Landsat 8 and Sentinel-2 (L + S) using four machine learning models.

As shown in Figure 6, the accuracy of the AGC model constructed based on RF is 0.94 and 0.62 in the modeling and validation phases, respectively, which are higher than the accuracy of the SVR model by 6.82 and 58.97%, respectively. In addition, the RMSEs decrease by 1.53 and 1.46 Mg/ha, and the overall modeling performance of RF is better than that of SVR. The accuracy of the AGC model constructed based on CatBoost is greater than that of XGBoost in both the modeling and validation phases, and the overall modeling effect of the CatBoost algorithm is slightly better than that of the XGBoost algorithm.

The modeling accuracy of the XGBoost algorithm is the same as that of RF, with $R^2$ being 0.94 for both, but the XGBoost validation accuracy is 8.06% higher than that of RF. The CatBoost algorithm outperforms SVR, RF, and XGBoost by 12.5, 5.32, and 5.32% at the modeling phase; and 79.49, 12.9, and 4.48% at the validation phase, respectively. The CatBoost algorithm has the highest accuracy according to the four AGC models constructed based on the characteristic variables of the L + S.

### 4.3. Spatiotemporal Distribution of AGC in the Shanghai Urban Forest

The model accuracy results of the urban forest AGC in Shanghai based on Landsat, Sentinel-2, and L + S data using four models, SVR, RF, XGBoost, and CatBoost, respectively, show that the AGC model based on L + S data using the CatBoost algorithm had the highest accuracy. Therefore, in this study, the prediction of urban forest carbon storage in Shanghai for different periods based on the characteristic variables of L + S using the CatBoost algorithm was carried out to obtain the spatial distribution of urban forest AGC in Shanghai from 2016 to 2019, and the results are shown in Figure 7.

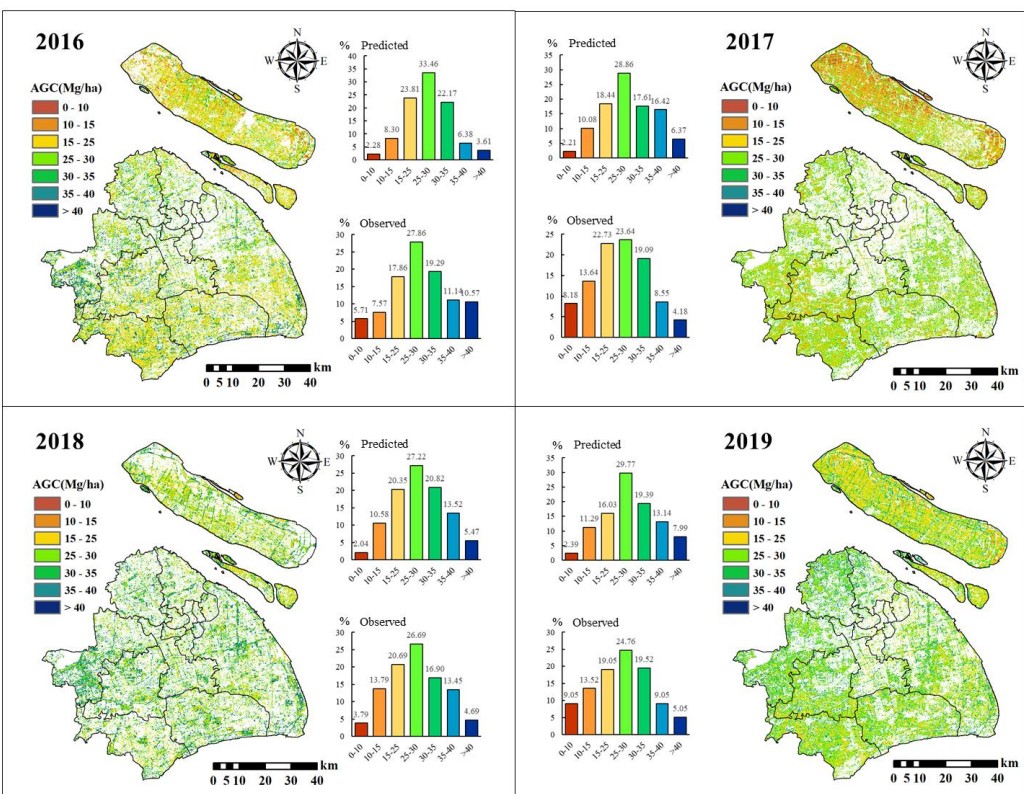

**Figure 7.** Prediction results of the spatiotemporal distribution of AGC in the Shanghai urban forest from 2016 to 2019.

The statistical analysis of Figure 7 shows that the average AGC of Shanghai forests from 2016 to 2019 was 24.90, 25.09, 25.17, and 25.61 Mg/ha, and the total carbon stocks were 2.37, 2.38, 2.40, and 2.35 Mt. There was no significant change in the spatial distribution of urban forest carbon stocks in Shanghai, with relatively small forest cover in the urban center. Far from the urban center, the western, eastern and northern regions showed lower carbon density but larger forest cover. This study further analyzed the frequency distribution histograms of the AGC of sample plots and model-estimated AGC. The results show that the model-inverted AGC histograms of all four phases of Shanghai forests were consistent with the structure of the sample plot survey, i.e., the largest proportion (27.22–33.46%) accounted for 20–25 Mg/ha, and the smaller proportion (2.04–2.39%) accounted for 0–10 Mg/ha. This indicates that the spatiotemporal distribution of the AGC estimated by the CatBoost model based on L + S data is consistent with the actual situation and can accurately reflect the spatial distribution characteristics of urban forest AGC in Shanghai.

## 5. Discussion

The input of different feature variables dramatically impacts the accuracy of AGC model construction. To understand the correlation between feature variables and AGC, this study conducted variable importance analysis on the feature variables of the three datasets screened using the Boruta method, and the ranking results are shown in Figure 8. The results show that the 11 feature variables based on the Landsat dataset are all texture information with different window sizes. The fifth band accounts for 54.55%, and 45.45% of the texture information is correlation features. Four of the five variables screened based on the Sentinel-2 dataset are texture features, among which 75% are correlation features; and 85% of the fourteen feature variables screened based on the L + S dataset are correlation features. This indicates that the critical factor in building forest AGC models is the texture information of Landsat, which is consistent with Zhang [10]. This suggests that the key factor in constructing forest AGC models is the texture information contained within

Landsat imagery, which is consistent with the results of Zhang's study. Meanwhile, 12 out of 14 feature variables screened based on the L + S dataset are Landsat texture information, and the accuracy of the constructed models is better than the previous two, which further verifies the importance of Landsat texture information in building forest AGC models. The urban forest has a complex structure, fragmented distribution, heterogeneous substratum, and frequent disturbances, which is very different from the large area and continuously distributed forests in traditional forest environments. Texture information is a crucial feature variable for constructing an urban forest AGC model. The acquisition of surface texture features, such as smoothness and fragility, from remotely sensed imagery allows for the identification of the subtle features of the urban forest.

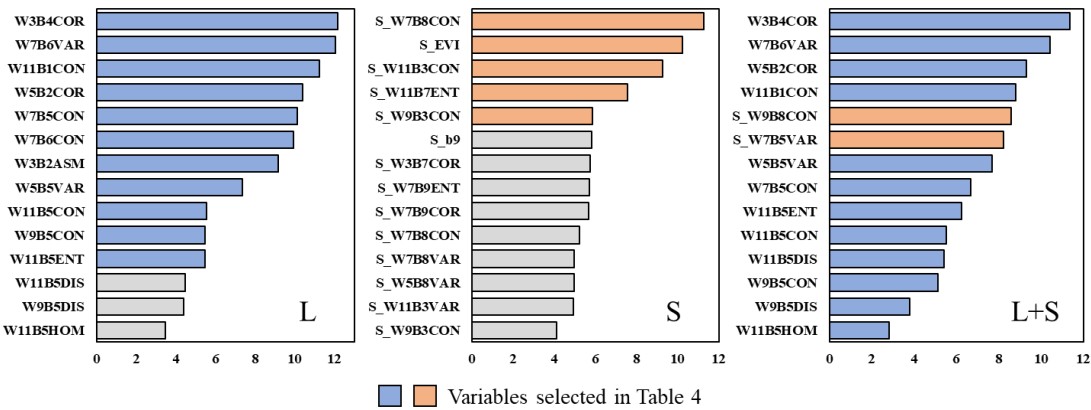

**Figure 8.** Variable importance ranking of Boruta for three datasets (L, S, L + S).

The traditional multiple linear regression (MLR) model is one of the most widely used methods for the remote sensing inversion of forest AGC [59–61]. In this study, MLR was also used to construct three data types for the Shanghai urban forest AGC estimation models, L, S, and L + S, as shown in Figure 9. The accuracy $R^2$ values of the MLR models constructed by L, S, and L + S are 0.26, 0.15, and 0.32, respectively, and the validation accuracy $R^2$ values are 0.36, 0.17, and 0.39, with an overall accuracy that is significantly lower than the four machine learning algorithms discussed in Section 4.2. The forest AGC spectral response tends to be nonlinear [62]; therefore, the MLR model cannot better explain the nonlinear characteristics between the independent and dependent variables.

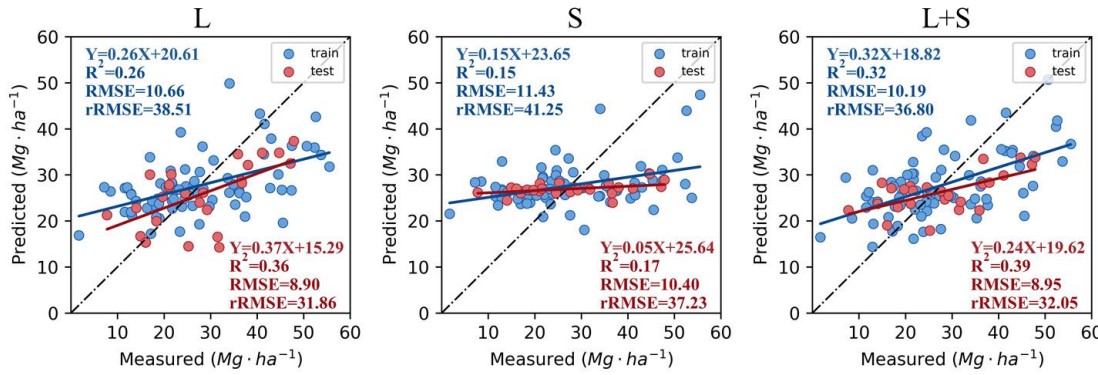

**Figure 9.** Accuracy evaluation of AGC prediction based on the MLR model.

In this study, the AGC modeling accuracy of XGBoost and CatBoost is higher than SVR and RF. Single machine learning algorithms have their own limitations. SVR solves the nonlinear problem by seeking hyperplanes, but the model itself needs to find the optimal kernel function and optimal penalty coefficients to obtain the optimal results. Although RF can handle high-dimensional data and has better noise immunity, it is prone to

overfitting problems when constructing models with a limited number of samples and many variables. XGBoost and CatBoost, as modern ensemble models, have better generalization and expression capabilities in densely distributed datasets and can automatically discover higher-order relationships between features and better describe the relationships between variables, thus improving the overall accuracy of the AGC model. In addition, CatBoost can better solve the problems of gradient bias and prediction shift; thus, the model performance is better than XGBoost in general.

Uncertainty analysis is the estimation and study of external factors and influences that cannot be controlled in the research process [63–65]. This study analyzed the uncertainty of the CatBoost model in estimating the forest AGC in Shanghai, as shown in Figure 10. Figure 10 shows that the percentage of image elements with low uncertainties (<20) in forest AGC for different periods from 2016 to 2019 is above 94%, indicating that the CatBoost model has good stability in estimating urban forest AGC in Shanghai, and the model is less influenced by external factors, further illustrating the accuracy of the forest AGC estimation results in this study. The remaining few image elements with large uncertainties (>20) were mainly distributed in the Chongming District of Shanghai. On the one hand, the forest AGC sample sites were unevenly distributed. The field survey plots' carbon stock data ranged from 1.71 to 87.10 Mg/ha, and the sample mean value of 28.59 Mg/ha was close to the overall sample mean, but its standard deviation of 19.12 Mg/ha was 1.57 times the overall sample standard deviation. This indicates that the sample data in the Chongming area have a large spatial heterogeneity, which may be impact the accuracy of estimating forest AGC with the CatBoost model. On the other hand, the estimation of urban forest AGC in Chongming District is relatively low (Figure 7), whereas the CatBoost model overestimates the low AGC values (Figure 6), which may be the reason for the great uncertainty in the estimation of forest AGC in Chongming District.

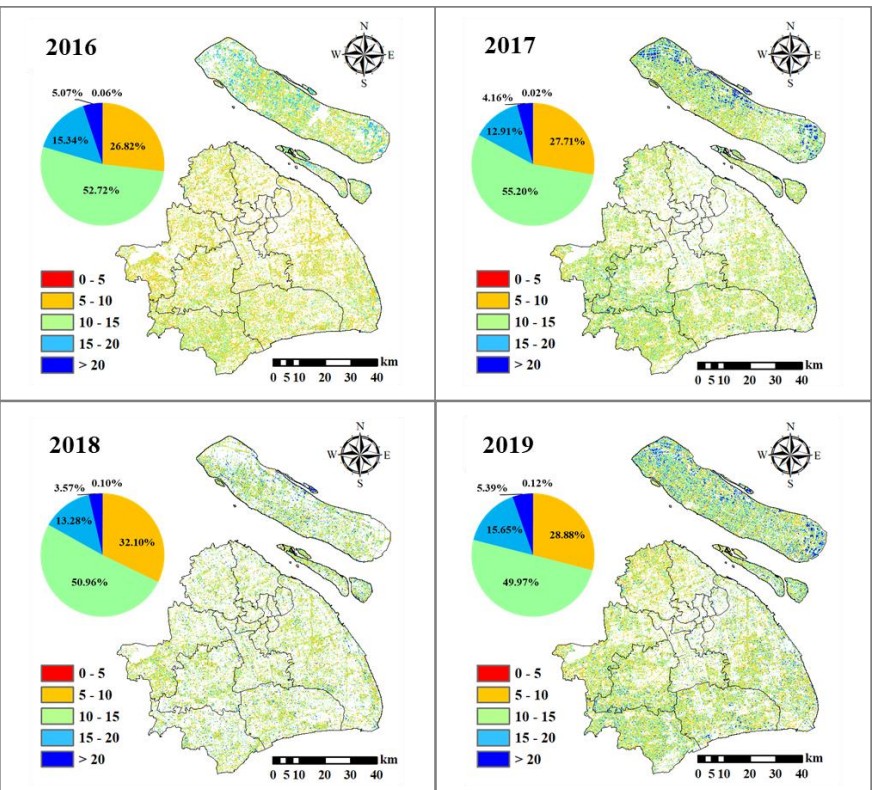

**Figure 10.** Uncertainty analysis of AGC prediction in Shanghai urban forest.

## 6. Conclusions

This study used four machine learning methods, SVR, RF, XGBoost, and CatBoost, to estimate the urban forest AGC in Shanghai based on three datasets, L, S, and L + S. The results show that: (1) The accuracy of the Shanghai urban forest AGC models based on L are all higher than that of S, whereas the model fitting accuracy and prediction accuracy of the models derived from L + S are the best overall. (2) Both types of machine learning models can achieve higher accuracy in predicting the AGC and spatiotemporal distribution of the Shanghai urban forest, but the AGC accuracy with the CatBoost model is relatively the highest, with fitting accuracy and validation accuracy $R^2$ values of 0.99 and 0.70, respectively, and RMSE values of 0.67 and 6.29 Mg/ha, respectively. The model prediction accuracy is smaller than that of SVR and RF by 79.49 and 12.9%, respectively, and RMSE by 25.56 and 10.01%, respectively. (3) The spatial distribution uncertainty of the AGC of the Shanghai urban forest obtained based on the CatBoost model prediction from 2016 to 2019 is small and consistent with the actual situation, and the statistics show that the AGC of the Shanghai forest is increasing, from 24.90 Mg/ha in 2016 to 25.61 Mg/ha in 2019. (4) Texture information is crucial for the construction of forest AGC models, and the results of the feature variables screened in this study based on three different datasets, Landsat, Sentinel-2, and L + S, all reflect the importance of texture information for the construction of urban forest AGC models in Shanghai.

**Author Contributions:** Conceptualization, H.D. and H.L.; data curation, G.Z., Q.Z. and L.X.; formal analysis, H.L.; funding acquisition, H.D.; investigation, G.Z., Q.Z. and L.X.; methodology, H.L. and H.D.; project administration, H.D.; software, H.L.; validation, H.L.; visualization, H.L.; writing original draft preparation, H.L.; writing review and editing, H.D. All authors have read and agreed to the published version of the manuscript.

**Funding:** The authors gratefully acknowledge the support of National Natural Science Foundation of China (U1809208, 32171785), the State Key Laboratory of Subtropical Silviculture (No. ZY20180201), and the Key Research and Development Program of Zhejiang Province (2021C02005).

**Data Availability Statement:** Landsat 8 OLI satellite data comes from Geospatial Data Cloud (http://www.gscloud.cn/ (accessed on 1 August 2021)), and Sentinel-2 satellite data comes from European Space Agency (https://scihub.copernicus.eu/ (accessed on 1 June 2022)).

**Acknowledgments:** The authors gratefully acknowledge the supports of various foundations. The authors are grateful to the editor and anonymous reviewers whose comments have contributed to improving the quality of this study.

**Conflicts of Interest:** The authors declare that they have no competing interest.

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
