# Peer review of "Prediction of Urban Forest Aboveground Carbon Using Machine Learning Based on Landsat 8 and Sentinel-2: A Case Study of Shanghai, China"

_remotesensing, doi:10.3390/rs15010284_

Round 1
Reviewer 1 Report
1- What is the purpose of this study? It should be briefly stated in the summary section
2- I would like to recommend some uptodate papers to make more strenght for Introduction section
1- Duysak, H. & YiÄŸit, E. Investigation of the performance of different wavelet-based fusions of SAR and optical images using Sentinel-1 and Sentinel-2 datasets . International Journal of Engineering and Geosciences , 7 (1) , 81-90 . DOI: 10.26833/ijeg.882589
2- Khorrami, B. & Valizadeh Kamran, K. (2022). A fuzzy multi-criteria decision-making approach for the assessment of forest health applying hyper spectral imageries: A case study from Ramsar forest, North of Iran . International Journal of Engineering and Geosciences , 7 (3) , 214-220 . DOI: 10.26833/ijeg.940166
3- Ahady, A. B. & Kaplan, G. (2022). Classification comparison of Landsat-8 and Sentinel-2 data in Google Earth Engine, study case of the city of Kabul . International Journal of Engineering and Geosciences , 7 (1) , 24-31 . DOI: 10.26833/ijeg.860077
4- Çömert, R. , Matcı, D. K. & Avdan, U. (2019). Object based burned area mapping with random forest algorithm . International Journal of Engineering and Geosciences , 4 (2) , 78-87 . DOI: 10.26833/ijeg.455595
5- Kaya, Y. & Polat, N. A linear approach for wheat yield prediction by using different spectral vegetation indices . International Journal of Engineering and Geosciences , 8 (1) , 52-62 . DOI: 10.26833/ijeg.1035037
33- Please explain data which you use at your study at Section 2,2 Please make a new section to explain data specifications. Sentinet-2 and landsat technical specifications should bi given at this section
34- Please explain shortly what is DVI , NDVI ,EVI, NDWI . some readers may not know these terms
Reviewer 2 Report
The authors report a comparison study on estimating urban forest aboveground carbon storage through the combination of satellite images and machine learning or ensemble learning methods. Although the novelty of the manuscript might be limited, it may provide guidance for future research in this topic. The manuscript is generally well written. I only have some comments for the authors to consider.
1. Line 40-45: Your study did not involve anything about natural forests, and I suggest you delete these sentences.
2. Line 59: How can measure DBH and tree height being destructive?
3. Line 77: Delete “both domestic and international”.
4. Line 146: More details on the field data are needed. Were the 81 plots surveyed in 2016 and 2017 spatially overlapped with those surveyed in 2018 and 2019? Moreover, how did you use them? Did you pair field data with remote sensing data based on time? This is a question for the modelling section.
5. Line 154: Even after reading the entire manuscript, I am still confused by why did you use both Landsat and Sentinel 2 images? They belong to the same category with very similar specifications, especially after you resampled the Sentinel2 images to a 30m resolution. Moreover, I am confused on your results that the accuracy of using Landsat is higher than that of using Sentinel 2. What are the potential reasons?
6. Line 182” You did not provide details on how you got the composite bands. Did you mean after band math or temporal fusion?
7. Line 217-219: Grammar issue.
8. Line 229: Random Forest is an ensemble learning method as well. I am confused by your division of machine learning and ensemble learning.
9. Line 340: All three paragraphs in this section are basically repeating the same sentences with different numbers. It would be better to rephrase them.
10. Line 436-438: Repeating sentence.
11. Line 462: I would like to see more explanations on the interpretation of the results in the Discussion section.
Reviewer 3 Report
see annexe
